# Effects of LED Lights with Defined Spectral Proportion on Growth and Reproduction of Indigenous Beijing-You Chickens

**DOI:** 10.3390/ani13040616

**Published:** 2023-02-09

**Authors:** Yanyan Sun, Yunlei Li, Shumei Ma, Lei Shi, Chao Chen, Dongli Li, Jiangpeng Guo, Hui Ma, Jingwei Yuan, Jilan Chen

**Affiliations:** 1Key Laboratory of Animal (Poultry) Genetics Breeding and Reproduction, Ministry of Agriculture and Rural Affairs, Institute of Animal Science, Chinese Academy of Agricultural Sciences, Beijing 100193, China; 2Pingliang Academy of Agricultural Sciences, Pingliang 744000, China; 3College of Animal Science and Technology, Hebei Agricultural University, Baoding 071001, China; 4Beijing Bainianliyuan Ecological Agriculture Co., Ltd., Beijing 101599, China; 5Beijing Innovation Consortium of Agriculture Research System, Beijing Animal Husbandry and Veterinary Station, Beijing 100101, China

**Keywords:** LED, light spectrum, laying hens, growth, reproduction

## Abstract

**Simple Summary:**

Vision is highly developed in fowl. Light not only provides ambient illumination, but also influences their physiological responses. Therefore, proper lighting management practices are crucial for improving the efficiency and welfare of commercial poultry operations. Light spectrum is the combination of different wavelengths of electromagnetic radiation produced by a lighting source. With a ban of traditional incandescent bulbs globally, the poultry industry is currently undergoing a shift to alternative lighting sources, such as light emitting diode (LED) bulbs, which permit the design of different and complex light spectrums output as required. The present study investigated the effects of three LED lights and one compact fluorescent lamp (CFL) with different spectrum characteristics on the performance during the pullet rearing and subsequent laying phase of Beijing-You chickens, a typical indigenous chicken breed in China. Results showed that the four lights did not show differences in feed efficiency before 17 week of age; the LED light with more balanced light spectrums (27% green light, 30% blue light, 22% yellow light, and 21% red light) promotes the prepubertal growth; and the LED light of more blue and green light (35% green light, 35% blue light, 18% yellow light, and 12% red light) provides proper sexual maturation age and better egg-laying persistence. These results inspire better design of LED lights for the indigenous chicken industry.

**Abstract:**

Light presents an important exogenous factor for poultry. This study examined effects of LED lights with different defined spectrums on growth and reproduction of indigenous Beijing-You chickens. A total of 576 one-day old female chicks were divided into 16 rooms, and each were exposed to four different lights: LED A (21% green light, 30% blue light, 24% yellow light, and 25% red light), B (35%, 35%, 18%, and 12%), C (27%, 30%, 22%, and 21%), or compact fluorescent lamps (CFL, 15%, 28%, 41%, and 16%). Results showed that feed intake and feed conversion ratio were comparable among treatments throughout the 17 week rearing period (*p* > 0.05). LED C showed similar body weight gain with CFL, but higher than LED A and B. The CFL birds start to lay on 132.25 d, while LED B did not lay until 148.25 d. The age at 50% egg production did not vary among groups (*p* = 0.12). Total egg number until 43 week of LED B was higher than others (*p* < 0.05). Therefore, LED lights with defined spectral proportion have different effects on chickens’ growth and reproduction. The LED C promotes the prepubertal growth, and the LED B provides proper sexual maturation age and better egg-laying persistence.

## 1. Introduction

Lighting is one of the most essential environmental factors for poultry production. It not only provides ambient illumination, but also influences animals’ physiological responses, behavior, growth, and reproduction [1]. With a ban of traditional incandescent bulbs, the poultry industry is undergoing a shift to alternative lighting sources, such as light-emitting diode (LED) bulbs and compact fluorescent lamps (CFL), which are luminous efficient and have long lifespans. Facets involved in light management of chickens include photoperiod, intensity, and wavelength (monochromatic light color) [2]. Extensive research has been conducted in the last few decades in broilers and laying hens to broaden the understanding of the separate or combined effects of these components, aiming at optimizing the lighting management to achieve maximum production efficiency and animal well-being.

Light spectrum is the combination of different wavelengths of electromagnetic radiation produced by a lighting source [3]. There is increasing attention in the literature with respect to LED spectrum reflecting the lighting quality as part of lighting programs [4]. It is shown that the different monochromatic light or different color temperature exert variable effects for the production efficiency [3,5,6,7]. In general, the monochromatic lights of short-wavelength (blue and green) reduce stress and promote growth, while the long-wavelength red monochromatic light promotes reproduction [8]. However, exposing birds to blue LED in the pullet phase and red LED in the egg-laying phase did not benefit overall egg production in the Hy-Line W-36 birds [9]. In view of combining the positive effect exerted by different artificial monochromatic lights, some studies involved the mixed monochromatic lights, which also showed their enhancements in animal performance [10]. However, the monochromatic lights may potentially impact the vision development in fowl and cause discomfort for the employee working in the poultry house. Full spectrum white LED bulbs may be designed to output different and complex light spectrums by adjusting the concentrations of different phosphors [11]. Furthermore, proper design of spectral proportion of LED as per varying breeds’ requirement and validation its practical outcome is therefore crucial [3,5]. Recently, Anette et al. (2021) designed LED lights outputting light spectrum similar to that of chicken’s ancestral forest environment and natural daylight, for which the white laying hens of the hybrid Bovans Robust showed higher preference and more active behavior [12]. Poultry specific LED products are emerging and have drawn increasing attention from both scientific and industry communities. Kamanli et al. (2015) indicated that the commercial laying hens from the incandescent, CFL, and cool daylight did not show difference in body weight (BW) at sexual maturity, feed intake (FI), feed conversion ratio (FCR), egg production, or egg qualities [13]. Long et al. (2016) demonstrated that there was no difference in egg weight, hen-day egg production, feed use, or mortality between DeKalb white laying hens housed under warm-white CFL and Nodark poultry-specific LED lights [14]. Liu et al. (2018) showed that the Hy-Line W-36 pullet reared under a dim-to-blue poultry-specific LED light had comparable performance of BW as those under the warm white CFL [15]. However, the studies are still lacking in local chicken breeds, for which their response to light spectrums may be different as compared to those high-productive ones due to the genetic difference.

The objectives of this study were to assess the effects of three designed LED lights with defined spectral proportion on growth, timing of sexual maturity, egg production performance, and egg quality of the Beijing-You chicken, an indigenous breed in China. The results are expected to contribute to the better design of lighting sources for various chicken breeds.

## 2. Materials and Methods

### 2.1. Animals and Experimental Design

This experiment involved four treatments: three LED bulbs with different light spectrums and one CFL. Their spectral power distribution was measured by the HP8000 Spectroradiometer System (HopTek Co., Ltd, Hangzhou, China), and shown in Figure 1 and Table 1. Different gradual peaks were seen in the LED lights. The LED A out spectrum is composed of 21% blue, 30% green, 24% yellow, and 25% red light, the LED B output spectrum is composed of 35% blue, 35% green, 18% yellow, and 12% red light, the LED C output spectrum is composed of 27% blue, 30% green, 22% yellow, and 21% red light. The CFL is characterized by many narrow peaks, and the out spectrum is composed of 15% blue, 28% green, 41% yellow, and 16% red light. Each treatment consisted of four light-proof replicate rooms containing 36 Beijing-You female chicks that are one day old (N = 576, average BW = 32.26 g, S.D. = 2.36 g) and housed in a pen (2.0 m × 1.2 m) at the beginning of this study. Beijing-you is a typical indigenous breed of dual purpose in China. The rooms were set up in an identical pattern, with the only difference being the lighting sources. The treatment last until 43 week of age. BW, FI, sexual maturity, and egg production traits were measured at the pen level. Other traits were measured from the random samples from the pen as detailed below.

### 2.2. Diets and Animal Husbandry

Diets were formulated based on a corn–soybean meal according to “The Feeding Standard of Chicken” (NY/T 33-2004) of Ministry of Agriculture and Rural Affairs of China. During the brooding period (0–6 week of age), the diet had 2.90 Mcal/kg metabolizable energy (ME), 20.00% of crude protein (CP), 0.90% of lysine (Lys), 0.38% of methionine (Met), and 0.90% of calcium (Ca). During the rearing period (7–17 week of age), the diet had 2.70 Mcal/kg ME and 15.00% of CP, 0.75% of Lys, 0.29% of Met, and 0.90% of Ca. From the 18 week of age to the onset of lay (5% egg production of each pen), the diet had 2.75 Mcal/kg ME and 16.00% of CP, 0.80% of Lys, 0.37% of Met, and 2.00% of Ca. During the laying period, the diet had 2.75 Mcal/kg ME and 16.0% of CP, 0.80% of Lys, 0.40% of Met, and 3.00% of Ca. Feed and water were provided ad libitum through the study. The lighting program was shown in Table 2. Light intensity was measured at the birds’ eye level using a light meter (model: DT-1301; Shenzhen Everbest Machinery Industry Co., Ltd., Shenzhen, China).

### 2.3. Growth Performance

The BW and FI were recorded at the end of 6, 12, and 17 week of age. The chickens were inspected thoroughly each day to record and any dead birds were removed. Mortality, body weight gain (BWG), FI, and FCR of 1–6 week, 7–12 week, 13–17 week, and 1–17 week were calculated accordingly.

### 2.4. Prepubertal Characteristics

At the end of 17 and 25 week of age, respectively, two birds from each replicate were randomly selected for weighing and processing. Birds were electrically stunned. The BW and eviscerated weights (without feather, blood and giblet weights) were recorded. The abdominal fat deposition is important for sexual maturation and, therefore, is also measured here. The abdominal fat pad including leaf fat surrounding the cloaca and gizzard were separated and weighed, and abdominal fat yield was calculated (abdominal fat weight/(abdominal fat weight + eviscerated weight) × 100%). The ovary weight was recorded, and ovary index was calculated (ovary weight/body weight, g/kg). At 25 week of age, the ovary was weighed and captured to count the hierarchal follicles number on the ovary.

### 2.5. Timing of Sexual Maturity

Age at the first egg and at 50% egg production was recorded for each replicate, and interval time between 5% and 50% egg production was calculated accordingly.

### 2.6. Egg Production Performance

During the laying period, daily egg production was recorded, and the egg-laying rate was calculated ((number of eggs/number of chickens) × 100%). Hen day egg production was calculated (total number of egg in 43 week period/average hen number per day). The following Yangning model [17] was used to fit the egg production curve:Y(t) = ae^−bt^/[(1 + e^−c(t−d)^)], 
in which, a is a scale parameter, b is the rate of decrease in laying ability, c is the reciprocal indicator of the variation in sexual maturity, and d is the mean age of sexual maturity of the hens.

### 2.7. Egg Quality

The egg quality parameters including egg weight, yolk weight, yolk percentage, egg shape index, and eggshell strength, which are important for the indigenous breed, were analyzed at 25 and 35 week of age, respectively, with 20 fresh eggs per pen measured at each age. The 20 eggs were randomly selected from the eggs collected in three consecutive days and analyzed within 24 h after the collection. The egg shape was measured by egg shape determinator (FHK, Fujihira Industry Co., Ltd., Tokyo, Japan) and the egg shape index were calculated (egg length/egg width). Eggshell strengths were measured by eggshell force gauge (FHK, Fujihira Industry Co., Ltd., Tokyo, Japan). The yolk was separated and weighed, and yolk percentage was calculated ((yolk weight/egg weight) × 100). Mean values of the 20 eggs of each pen were calculated for the subsequent statistical analysis.

### 2.8. Antioxidant Status and Lipid Peroxidation

At 17 week of age, 2.0 mL of blood sample was drawn from three birds randomly selected from each replicate by venipuncture with coagulation-promoting vacuum tubes. Then, blood samples were centrifuged at 3000× *g* for 5 min at 4 °C. The upper layer was collected and stored at −80 °C. Serum concentration of malondialdehyde (MDA), superoxide dismutase (SOD), glutathione peroxidase (GSH-PX), total antioxidation (T-AOC), and melatonin (MT) were assayed by using colorimetric methods with commercial kits (Nanjing Jiancheng Bioengineering Institute, Nanjing, China). Each sample was assayed in three replicates.

### 2.9. Statistical Analysis

Data were analyzed using one-way ANOVA (SAS 9.1, SAS Institute Inc., Cary, NC, USA). Significance was designated as *p* < 0.05. Means were compared by Student–Newman–Keuls multiple-range tests when a significant difference was detected.

## 3. Results

This section may be divided by subheadings. It should provide a concise and precise description of the experimental results, their interpretation, and the experimental conclusions that can be drawn.

### 3.1. Growth Performance of Pullets

As illustrated in Table 3, FI, and FCR were comparable for all groups at any stages (week 1–6, 7–12, 13–17, and 1–17) throughout the 17-week prepubertal rearing period (*p* > 0.05). The BWG during week 1–6 (*p* = 0.01), 13–17 (*p* = 0.03), and 1–17 (*p* = 0.02) differs among the four groups. In the first 6 weeks, the birds of the LED C group had higher BWG than the LED A and B (*p* < 0.05). While from 13 to 17 week, the birds of the LED B and C group had higher BWG than the LED A group (*p* < 0.05). Furthermore, the relative trend was consistent with time. For the whole rearing period, the LED C group showed higher BWG than the LED A and B groups, and comparable with that of the CFL group. The cumulative mortality of pullets under LED B was higher than those under LED C and CFL (*p* < 0.05).

### 3.2. Evaluation of Antioxidant Status and Lipid Peroxidation

Table 4 showed the serum concentration of MDA, SOD, GSH-Px, T-AOC, and MT of the birds at 17 week of age under four lighting sources. The SOD of LED C was comparable with CFL (*p* > 0.05), and higher than LED A and LED B (*p* < 0.05). The T-AOC of LED A was higher than other three groups (*p* < 0.05). The GSH-Px of LED B and LED C was higher than LED A and CFL (*p* < 0.05). The four groups did not show difference in MDA (*p* = 0.49) or MT (*p* = 0.20) concentration.

### 3.3. Peripubertal Characteristics

Peripubertal characteristics including BW, sexual organ development, abdominal fat deposition, etc. at 17 and 25 week of age were presented in Table 5. At 17 week of age, BW (*p* < 0.0001), eviscerated weight (*p* = 0.04), ovary weight (*p* = 0.007) and index (*p* = 0.02), and abdominal fat weight (*p* = 0.004) and yield (*p* = 0.01) were different between groups. The BW, ovary weight, and abdominal weight of LED B were lower than the other three groups (*p* < 0.05). The ovary index and abdominal fat yield of LED B were lower than LED A and CFL (*p* < 0.05). At 25 week of age, BW (*p* = 0.82), ovary weight (*p* = 0.79) and index (*p* = 0.84), follicles number (*p* = 0.99), and abdominal fat weight (*p* = 0.32) and yield (*p* = 0.15) were comparable among groups.

### 3.4. Sexual Maturity and Egg Production Performance

The age of the first egg and 50% egg production of Beijing-You chickens reared under different lighting sources are shown in Table 6. A significant delay in laying was observed when the chickens were reared under LED A, LED B, and LED C compared to CFL (*p* = 0.004). The birds in CFL start to lay as early as 132.25 d, while those in LED B did not start to lay until 148.25 d. The age of 50% egg production did not vary among groups (*p* = 0.12). The time interval between the first egg to 50% egg production ranged from 20.50 d in LED A to 38.00 d in CFL, and the four groups differ (*p* = 0.03). Hen day egg production (43 week) of LED B was higher than other three groups (*p* < 0.05).

The actual and fitted egg-laying rate curves of each group are shown in Figure 2. The models for each group are as follows:$$\mathrm{LED}\;A:\;y\left(t\right)=78.8476\times e^{-0.00801\times t}/\left[\left(1+e^{-0.6909\times \left(t-23.7346\right)}\right)\right],\;R^{2}=0.9667,$$
$$\mathrm{LED}\;B:\;y\left(t\right)=67.2803\times e^{-0.00227\times t}/\left[\left(1+e^{-0.7289\times \left(t-24.5798\right)}\right)\right],\;R^{2}=0.9979,$$
$$\mathrm{LED}\;C:\;y\left(t\right)=89.9666\times e^{-0.01210\times t}/\left[\left(1+e^{-0.5597\times \left(t-23.9244\right)}\right)\right],\;R^{2}=0.9975,$$
$$\mathrm{CFL}:\;y\left(t\right)=77.2783\times e^{-0.00584\times t}/\left[\left(1+e^{-0.4621\times \left(t-24.6231\right)}\right)\right],\;R^{2}=0.9977.$$

The estimated parameters in the models illustrate the egg-laying characteristics of each group. As the rate of decrease in laying ability, the b of LED C was high. As the reciprocal indicator of the variation in sexual maturity, the c of the CFL group was low. As the mean age of sexual maturity of the hens, the d of LED A was the least, and that of LED B group was the largest.

### 3.5. Egg Qualities

As shown in Table 7, the egg weight, yolk weight, yolk percentage, egg shape index, and eggshell strength of both 25 and 35 week of ages were in normal range and did not differ between four groups (*p* > 0.05).

## 4. Discussion

Proper lighting management practices are crucial to improving the efficiency, output, and welfare of commercial poultry operations. While there has been considerable research on the other factors, there has been relatively limited investigation on the spectrum, especially for indigenous chicken breeds. The present study investigated the effects of three LED lights and one CFL light with different spectrum characteristics on the performance during the pullet rearing and subsequent laying phase of Beijing-You, a typical indigenous chicken breed.

### 4.1. Effects of Lighting Sources on Growth Performance of Pullets

The green, blue, white, red, and yellow light are the most studied monochromatic lights used in poultry lighting management studies. It was reported that the blue and green lights may improve the growth of broilers as compared with white and red lights [18], as these monochromatic lights were more effective to alleviate the stress response [19] and enhance immune response [20,21], decrease oxidative stress [22], promote stimulate satellite cell myogenic processes [23], and stimulate testosterone secretion and myofiber growth that led to increased body growth [24]. The combination of blue and green monochromatic lights by switching at critical points between the early and later stages of growth exhibited better effects for broilers [24]. No difference was found for broiler growth when rearing under different LED lights and typical CFL light [14,25]. In the laying hens, Baxter et al. (2019) reported that pullets reared under red, green, or white LED light had comparable BW until 23 week of age [26]. Liu et al. (2018) indicated that the pullets under dim-to-blue LED light and CFL light had comparable BW throughout the rearing period and proposed that the impact of spectral characteristics of the lighting sources might be negligible on the growth performance of pullets [15]. Takeshima et al. (2019) indicated there was no consistent significant difference in BW between the pullets exposed to either 50% red LED or 60% green LED [27]. The three LED lights and the CFL lights used in the present study had distinctly different spectral characteristics. The LED B light possess higher blue and green light spectrum. However, the pullet under the LED B light showed the lowest BWG. This difference in pullets and broilers may be attribute to the fact that they may have different growth responses to light [15].

### 4.2. Effects of Lighting Sources on Anti-Oxidative Status

The MDA, SOD, GSH-Px, and T-AOC concentration are the classical parameters employed to evaluate the antioxidant status and lipid peroxidation of livestock. MT also has antioxidant effects exerting through direct or indirect mechanisms [28]. Their circulation concentration may be associated with animal performance, and could be affected by the lighting spectrum. Ke et al. (2011) revealed that blue monochromatic light elevated T-AOC, SOD, and GSH-Px activity, and reduced MDA contents in the muscles as compared with red and white light [22]. Jin et al. (2011) [29] and Li et al. (2014) [30] reported in broilers that the green light could enhance the pinealocyte to secrete MT. However, in this study, these parameters responded differently to the lighting sources. The GSH-Px of the LED B group was higher. It is known that LED B was composed of the highest proportion of blue (35%) and green light (35%), which were reported to be able to elevate the GSH [31]. The T-AOC is composed of enzymatic and non-enzymatic antioxidant defense systems, reflecting the body’s total antioxidant and free radical scavenging capacity. The decreased SOD and GSH-Px, but increased level of T-AOC in the LED A group may suggest that the ability to use non-enzymatic antioxidant substances was enhanced in this group.

### 4.3. Effects of Lighting Sources on Sex Maturation and Reproduction Performance

The laying hen response to the photoperiodic cues to initiate sexual maturation via the activation of the hypothalamo-pituitary-gonadal axis [32,33]. Red light is reported as a more effective stimulatory wavelength than green light required to stimulate the reproductive axis [34,35]. Wei et al. (2020) indicated that the birds under yellow-orange light during the brooding period of 1 to 20 week arrived 50% egg production earlier than the those under white and blue-green light [5]. The birds in the CFL group start to lay the earliest, and those of the LED B group were the last to lay. The LED B light consists of the most blue and green light spectrum. Therefore, this observation seems consistent with the proposed rules from the literatures that the green light may have an inhibitory effect on reproductive performance and longer wavelength may stimulate sexual maturation. However, it is interesting that the hen day egg number until 43 week of age of the LED B group was higher than other three groups. In the present study, the photoperiod and light intensity increased from 18 week of age for all groups. The reproductive system of the laying hen changes from an immature nonfunctional state to a mature functional state during the 2- to 4 week period following photostimulation [36]. The early age of the first egg did not promise longer lasting high egg production. Instead, the proper sexual maturation age is crucial for producing the maximum egg mass [37,38]. It is also noted that the BW, ovary weight, and abdominal fat deposition were lower in the LED B group than other three groups. The body maturation is the prerequisite for sexual maturation. It was possible that the LED B light provided the Beijing-You chickens the proper sexual maturation age and, therefore, the better egg-laying persistence.

### 4.4. Effects of Lighting Sources on Egg Qualities

In the present study, some important egg quality parameters for indigenous chicken breeds are estimated to determine the potential lighting sources’ effect. Indigenous chicken breeds are prone to produce eggs of smaller size, longer shape, and with higher yolk proportion than commercial laying hens. Although genetically determined [39], these traits were also varied with environmental factors including light stimulation. Li et al. (2014) indicated that hens under red monochromatic light (660 nm) produced heavier and longer eggs than those under the incandescent light, blue (480 nm) or green (560 nm) monochromatic light, while their egg production was relatively lower [40]. The LED light was reported to result in higher egg weight, albumen height, and albumen weight at 27 week of age, thicker shells at 40 week of age, but lower egg weight at 60 week of age as compared to the CFL [14]. On the other hand, there are also a bunch of studies showing no difference among light treatments in egg quality traits [41,42]. Similarly in the current study, the three LED lighting sources and the CFL led to comparable egg quality traits in terms of egg weight, yolk proportion, and eggshell strength.

## 5. Conclusions

This study examined the effects of three designed LED lights with different defined spectral proportion and one CFL on growth, timing of sexual maturity, egg production performance, and egg quality of an indigenous chicken breed. In general, the results indicate that as compared to CFL, LED lights exert more significant effect on prepubertal body and sexual maturation and the overall egg number. More specifically, the LED C was shown to promote the prepubertal growth and sexual maturation, while the LED B provide the proper sexual maturation age and, therefore, better egg-laying persistence. These results inspire better design of LED light resources for chickens.

## Figures and Tables

**Figure 1 animals-13-00616-f001:**
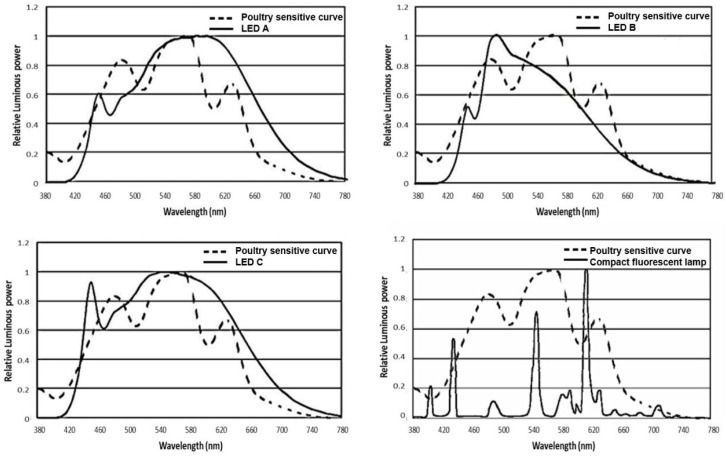
Comparison of spectrum readings of the three light emitting diode bulbs and one compact fluorescent lamp used in this study. LED, light emitting diode; The poultry sensitive curve was adapted from a previous publication [16].

**Figure 2 animals-13-00616-f002:**
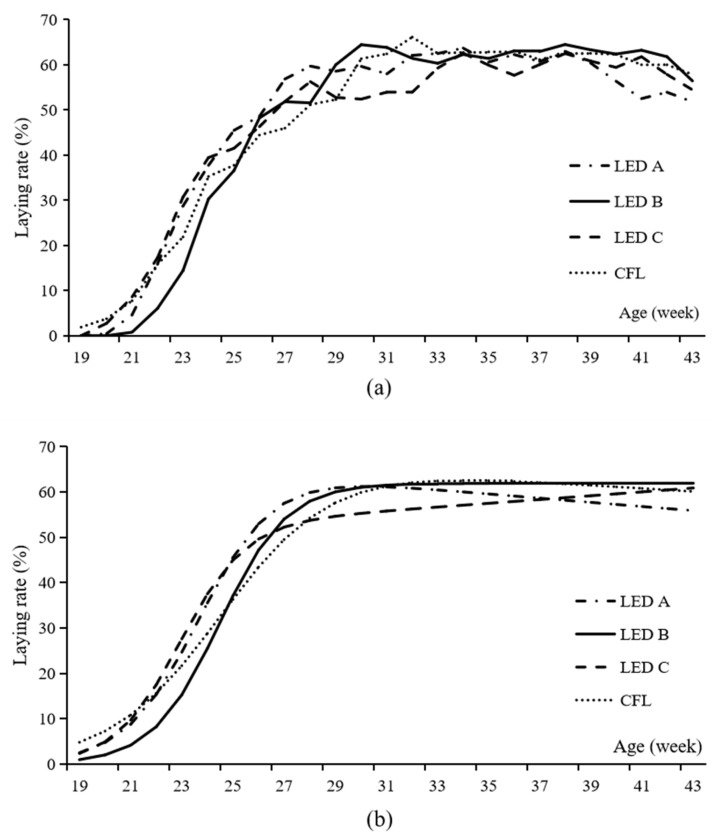
The actual (**a**) and fitted (**b**) laying rate cure of Beijing-You chickens reared under three light emitting diode bulbs and one compact fluorescent lamp. LED, light emitting diode; CFL, compact fluorescent lamp. The out spectrum of blue green, yellow, and red light is 21%, 30%, 24%, and 25% for LED A; 35%, 35%, 18%, and 12% for LED B; 27%, 30%, 22%, and 21% for LED C; and 15%, 28%, 41%, and 16% for CFL, respectively.

**Table 1 animals-13-00616-t001:** Spectrum proportion characteristics of three light emitting diode bulbs and one compact fluorescent lamp.

Lighting Sources ^1^	Blue Light	Green Light	Yellow Light	Red Light
LED A	21%	30%	24%	25%
LED B	35%	35%	18%	12%
LED C	27%	30%	22%	21%
CFL	15%	28%	41%	16%

^1^ LED, light emitting diode; CFL, compact fluorescent lamp.

**Table 2 animals-13-00616-t002:** Lighting program of Beijing-You chickens used in the present study.

Age (Week)	Photoperiod (h/d)	Lighting Intensity (lux)
1	24	20
2	22	10
3	20	10
4	18	10
5	16	10
6	14	10
7	12	10
8	10	10
9–17	9	10
18	10	20
19	11	20
20	12	20
21	13	20
22	14	20
23	15	20
24–43	16	20

**Table 3 animals-13-00616-t003:** Effect of lighting sources on growth performance of Beijing-You pullets.

Age	Trait ^1^	Lighting Source ^2^				SEM	*p*-Value
		LED A	LED B	LED C	CFL		
1–6 week	FI (g/bird)	1010.95	1069.80	1018.40	1098.99	38.89	0.36
	BWG (g/bird)	330.61 ^b^	316.40 ^b^	350.49 ^a^	333.84 ^ab^	5.74	0.01
	FCR	3.06	3.38	2.91	3.30	0.13	0.08
	Mortality (%)	1.39	2.78	0.00	2.78	1.33	0.43
7–12 week	FI (g/bird)	2280.68	2068.52	2094.41	1926.44	186.42	0.62
	BWG (g/bird)	603.46	546.61	612.28	630.49	27.83	0.22
	FCR	3.79	3.75	3.42	3.06	0.24	0.16
	Mortality (%)	2.82	6.29	2.08	1.39	1.53	0.16
13–17 week	FI (g/bird)	2236.93	2197.63	2506.68	2486.74	124.45	0.22
	BWG (g/bird)	310.76 ^b^	359.80 ^a^	360.04 ^a^	333.75 ^ab^	11.36	0.03
	FCR	7.21	6.18	6.97	7.48	0.46	0.26
	Mortality (%)	1.43	0.76	0.00	0.00	0.56	0.26
1–17 week	FI (g/bird)	5528.56	5335.95	5619.49	5512.17	211.80	0.82
	BWG (g/bird)	1244.83 ^bc^	1222.81 ^c^	1322.81 ^a^	1298.08 ^ab^	20.69	0.02
	FCR	4.44	4.35	4.25	4.25	0.14	0.73
	Mortality (%)	5.56 ^ab^	8.33 ^a^	2.08 ^b^	2.28 ^b^	1.43	0.04

^a,b,c^ Means in the same column within the same treatment factor with different small letter superscripts indicate *p* < 0.05; ^1^ FI, feed intake; BWG, body weight gain; FCR, feed conversion ratio; ^2^ LED**,** light emitting diode; CFL, compact fluorescent lamp. The out spectrum of blue green, yellow, and red light is 21%, 30%, 24%, and 25% for LED A; 35%, 35%, 18%, and 12% for LED B; 27%, 30%, 22%, and 21% for LED C; and 15%, 28%, 41%, and 16% for CFL, respectively.

**Table 4 animals-13-00616-t004:** Effect of lighting sources on of serum antioxidant status and lipid peroxidation of Beijing-You chickens at 17 week of age.

Trait ^1^	Lighting Source ^2^				SEM	*p*-Value
	LED A	LED B	LED C	CFL		
MDA (nmol/mL)	9.73	9.82	9.64	9.76	0.09	0.49
SOD (U/mL)	83.96 ^b^	78.49 ^b^	98.28 ^a^	107.39 ^a^	4.92	0.001
GSH-P_X_ (μmol/L)	620.90 ^b^	699.17 ^a^	683.20 ^a^	602.91 ^b^	10.57	<0.0001
T-AOC (U/mL)	34.72 ^a^	30.71 ^b^	29.79 ^b^	31.33 ^b^	0.89	0.002
MT (ng/L)	135.40	137.93	130.98	162.09	10.97	0.20

^a,b^ Means in the same column within the same treatment factor with different small letter superscripts indicate *p* < 0.05. ^1^ MDA, malondialdehyde; SOD, superoxide dismutase; GSH-Px, glutathione peroxidase; T-AOC, total antioxidant capability; and MT, melatonin; ^2^ LED, light emitting diode; CFL, compact fluorescent lamp. The out spectrum of blue green, yellow, and red light is 21%, 30%, 24%, and 25% for LED A, 35%, 35%, 18%, and 12% for LED B, 27%, 30%, 22%, and 21% for LED C, and 15%, 28%, 41%, and 16% for CFL, respectively.

**Table 5 animals-13-00616-t005:** Effect of lighting sources on prepubertal characteristics of Beijing-You chickens.

Age	Trait	Lighting Source ^1^				SEM	*p*-Value
		LED A	LED B	LED C	CFL		
17 week	BW (kg)	1.28 ^b^	1.25 ^c^	1.34 ^a^	1.35 ^a^	0.01	<0.0001
	Eviscerated weights (kg)	0.91 ^b^	0.90 ^b^	0.97 ^a^	0.96 ^a^	0.01	0.04
	Ovary weight (g)	0.95 ^a^	0.72 ^b^	0.87 ^a^	0.97 ^a^	0.05	0.007
	Ovary index (g/kg)	0.74 ^a^	0.58 ^b^	0.65 ^ab^	0.72 ^a^	0.04	0.02
	Abdominal fat (g)	53.82 ^a^	36.03 ^b^	51.54 ^a^	59.00 ^a^	4.39	0.004
	Abdominal fat yield (%)	5.56 ^a^	3.84 ^b^	5.04 ^ab^	5.80 ^a^	0.43	0.01
25 week	BW (kg)	1.75	1.80	1.76	1.82	0.06	0.82
	Eviscerated weights (kg)	1.21	1.22	1.19	1.23	0.12	0.67
	Ovary weight (g)	30.23	35.22	31.53	38.38	6.27	0.79
	Ovary index (g/kg)	17.01	18.98	17.96	21.21	0.45	0.84
	Hierarchal follicles number	4.91	5.18	4.92	5.25	0.87	0.99
	Abdominal fat (g)	87.55	104.42	103.58	77.00	12.13	0.32
	Abdominal fat yield (%)	6.74	7.44	7.87	5.80	0.66	0.15

^a,b,c^ Means in the same column within the same treatment factor with different small letter superscripts indicate *p* < 0.05. ^1^ LED, light emitting diode; CFL, compact fluorescent lamp. The out spectrum of blue green, yellow, and red light is 21%, 30%, 24%, and 25% for LED A; 35%, 35%, 18%, and 12% for LED B; 27%, 30%, 22%, and 21% for LED C; and 15%, 28%, 41%, and 16% for CFL, respectively.

**Table 6 animals-13-00616-t006:** Effect of lighting sources on egg production of Beijing-You chickens.

Trait	Lighting Source ^1^				SEM	*p*-Value
	LED A	LED B	LED C	CFL		
Age at first egg (d)	144.25 ^ab^	148.25 ^a^	140.25 ^b^	132.25 ^c^	2.44	0.004
Age at 50% egg production (d)	164.75	173.25	163.00	170.25	3.11	0.12
Interval from first egg to 50% production (d)	20.50 ^b^	25.00 ^b^	22.75 ^b^	38.00 ^a^	3.81	0.03
Hen-day egg number until 43 week	80.55 ^b^	92.09 ^a^	78.91 ^b^	79.76 ^b^	3.13	0.04

^a,b,c^ Means in the same column within the same treatment factor with different small letter superscripts indicate *p* < 0.05. ^1^ LED**,** light emitting diode; CFL, compact fluorescent lamp.

**Table 7 animals-13-00616-t007:** Effect of lighting sources on egg quality traits of Beijing-You chickens at 25 and 35 week of age.

Age	Trait	Lighting Source ^1^				SEM	*p*-Value
		LED A	LED B	LED C	CFL		
25 week	Egg weight (g)	40.42	41.22	40.86	40.70	0.95	0.93
	Yolk weight (g)	11.85	12.34	11.75	12.03	0.41	0.78
	Yolk percentage (%)	29.6	30.01	28.79	28.59	0.48	0.17
	Egg shape index	1.27	1.28	1.26	1.30	0.01	0.22
	Eggshell strength (kg/cm^2^)	3.28	3.33	3.15	3.44	0.11	0.68
35 week	Egg weight (g)	49.46	49.98	49.66	50.31	0.78	0.88
	Yolk weight (g)	16.28	16.25	15.69	16.69	0.28	0.15
	Yolk percentage (%)	32.99	32.52	31.58	32.93	0.41	0.11
	Egg shape index	1.31	1.32	1.32	1.33	0.01	0.85
	Eggshell strength (kg/cm^2^)	3.64	3.85	3.64	3.64	0.16	0.71

^1^ LED, light emitting diode; CFL, compact fluorescent lamp.

## Data Availability

Not applicable.

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
