# Peer review of "Effects of LED Lights with Defined Spectral Proportion on Growth and Reproduction of Indigenous Beijing-You Chickens"

_animals, 2023, doi:10.3390/ani13040616_

Round 1

Reviewer 1 Report

This manuscript descibed a significant effect of LED lights on the growth and the reproductive performance of the Beijing-You chicken.  Although the results showed promising merit of the use of LED lights, yet the benefit gained from the use of LED lights seems not too great.  A few points are addressed as following:

1. Line 29-31: authors showed the percentage of gree, blue, yellow and red lights for LED C and LED B, but did not provide the rationale of each light source.

2.Two times used the wrong word for "feed conversion rate".  First time in line38, "feed conservation rate" was used.  Second time in line 77, "feed conversation rate" was used.

3. Line 94: CFL was used as a control.  What makes CFL be a control?

4. Line 175: The statistics should be a separate paragraph, with 2.9 as title.

Author Response

Response to Reviewer 1 Comments

This manuscript described a significant effect of LED lights on the growth and the reproductive performance of the Beijing-You chicken.  Although the results showed promising merit of the use of LED lights, yet the benefit gained from the use of LED lights seems not too great. A few points are addressed as following:

1. Line 29-31: authors showed the percentage of green, blue, yellow and red lights for LED C and LED B, but did not provide the rationale of each light source.

Re: The LED C was with more balanced light spectrums, while LED B output higher proportion of blue and green light. This information was added in line 30-32 in the revised manuscript.

2. Two times used the wrong word for "feed conversion rate".  First time in line 38, "feed conservation rate" was used.  Second time in line 77, "feed conversation rate" was used.

Re: These errors were corrected. “Feed conversion ratio” was used in line 41, 85, and 204 in the revised manuscript.

3. Line 94: CFL was used as a control. What makes CFL be a control?

Re: It may be improper by saying so. “as a control” was removed from this paragraph in the revised manuscript.

4. Line 175: The statistics should be a separate paragraph, with 2.9 as title.

Re: The “2.9 Statistical Analysis” was added as a title in the revised manuscript. 

Reviewer 2 Report

his study compared the effects of three LED lights with defined spectral proportion on growth and reproduction traits of a local chicken breed. It is relevant both to the science and industry aspect, because the poultry industry is switching the lighting device to LED light. Here are some minor comments for the authors to consider to improve the description of this study.

1.    LINE 32 Spell error for THESES

2.    LINE 123. It is stated that the diet changed from 5% egg laying rate, did this 5% based on the animals in each pen or based on the all animals? It should be clear.

3.    What is the mortality during the rearing period of each group? It is also an important index may be affected by the light environment.

4.    LINE 149. Should “-bt” in the Yangning Model is superscript to “e”, as it is different in the fitted models in line 234-237.

5.    LINE 228. Hen day egg production calculation should be defined somewhere.

6.    Figure 2. The spectrum proportion characteristics of three LED lights and compact fluorescent lamp in Table 1 should be added in the footnote of Figure 2. Otherwise the reader have no idea about the difference of the LED A B C based merely on the figure itself.

Author Response

Response to Reviewer 2 Comments

This study compared the effects of three LED lights with defined spectral proportion on growth and reproduction traits of a local chicken breed. It is relevant both to the science and industry aspect, because the poultry industry is switching the lighting device to LED light. Here are some minor comments for the authors to consider to improve the description of this study.

1. LINE 32 Spell error for THESES

Re: “These” was used to replace “These”. Please see line 34 in the revised manuscript.

2. LINE 123. It is stated that the diet changed from 5% egg laying rate, did this 5% based on the animals in each pen or based on the all animals? It should be clear.

Re: It was based on the pen level. This information was added in line 131 in the revised manuscript.

3. What is the mortality during the rearing period of each group? It is also an important index may be affected by the light environment.

Re: The overall mortality during the rearing period of each group was added in Table 3.

4. LINE 149. Should “-bt” in the Yangning Model is superscript to “e”, as it is different in the fitted models in line 234-237.

Re: Yes. The model was checked and modified for this error. Please see line 161 in the revised manuscript.

5. LINE 228. Hen day egg production calculation should be defined somewhere.

Re: The definition was added in line 158-159 in the revised manuscript.

6. Figure 2. The spectrum proportion characteristics of three LED lights and compact fluorescent lamp in Table 1 should be added in the footnote of Figure 2. Otherwise the reader have no idea about the difference of the LED A B C based merely on the figure itself.

Re: This information was added for Figure 2 and other relevant tables in the revised manuscript.

Reviewer 3 Report

General comments:

The authors aimed to investigate the effects of 33 LED lights with different defined spectrum on growth and reproduction of indigenous Beijing-You 34 chickens.

 1.     The significance of the author's research is insufficient, and more literatures need to be retrieved to support the purpose and significance of this article.

2.     There are numerous errors in grammar, selection of words, and expressions in the manuscript. A careful correction of the entire manuscript is needed. Please polish it thoroughly by the English native speaker.Comment related to specific sections and locations:

1.     Line 94, A comparison of spectra between these bulbs was Their spectral power distribution was characteristics were measured by the HP8000 Spectroradiometer System (HopTek Co. LTD, Hangzhou, China) and shown in Figure 1 and Table 1. was Their ?

2.     Line 95, Their spectral power distribution was characteristics were measured by the HP8000 Spectroradiometer System. The “was” should be delete?

3.     Line 117, to “Feeding Standard……” should be to the Feeding…….Please modify the manuscript.

Author Response

Response to Reviewer 3 Comments

The authors aimed to investigate the effects of LED lights with different defined spectrum on growth and reproduction of indigenous Beijing-You chickens.

1. The significance of the author's research is insufficient, and more literatures need to be retrieved to support the purpose and significance of this article.

Re: We added four reference to support the purpose and significance of this article. Please see line 64, 68-70, 78-81, and the reference list in the revised manuscript.

2. There are numerous errors in grammar, selection of words, and expressions in the manuscript. A careful correction of the entire manuscript is needed. Please polish it thoroughly by the English native speaker.

Re: Thanks for your suggestion. We have invited a native English speaker to polish the manuscript.

Comment related to specific sections and locations:

1. Line 94, A comparison of spectra between these bulbs was Their spectral power distribution was characteristics were measured by the HP8000 Spectroradiometer System (HopTek Co. LTD, Hangzhou, China) and shown in Figure 1 and Table 1. was Their?

Re: This sentence was modified as follows.

“Their spectral power distribution was measured by the HP8000 Spectroradiometer System (HopTek Co. LTD, Hangzhou, China) and shown in Figure 1 and Table 1”

2. Line 95, Their spectral power distribution was characteristics were measured by the HP8000 Spectroradiometer System. The “was” should be delete?

Re: This sentence was modified as follows.

“Their spectral power distribution was measured by the HP8000 Spectroradiometer System (HopTek Co. LTD, Hangzhou, China) and shown in Figure 1 and Table 1”

3. Line 117, to “Feeding Standard……” should be to the Feeding…….Please modify the manuscript.

Re: It was modified as suggested.

Reviewer 4 Report

Line 25-26 - all abbreviations must be explained when 1st time appera

In Introduction or M&M 2 sentences should be added motivating the material choice, also some characteristics of chosen breed should be described.

the experimental factor was applied only during the prepubertal period? or during the whole birds' rearing? It must be clearly stated

Line 103-104 - the age of birds at the beginning of the experiment should be given

Line 195 - BWG - incorrect explanantion (body weight gain)

Line 219 - it is not necessary to repeat formulas mentioned before in M&M

How the terms of eggs quality evealuation were chosen?

What was the duration of experiment?

Author Response

Response to Reviewer 4 Comments

1. Line 25-26 - all abbreviations must be explained when 1st time appera

Re: The abbreviation of LED and CFL was defined as suggested. Please see line 25-26 in the revised manuscript.

2. In Introduction or M&M 2 sentences should be added motivating the material choice, also some characteristics of chosen breed should be described.

Re: Thank you for your suggestion. These sentences were added in line 90-92, 95 in the revised manuscript.

3. The experimental factor was applied only during the prepubertal period? or during the whole birds' rearing? It must be clearly stated

Re: The experimental factor was applied during the whole birds’ rearing. This information was added in line 112-113 in the revised manuscript.

4. Line 103-104 - the age of birds at the beginning of the experiment should be given

Re: This information was added in line 109 in the revised manuscript.

5. Line 195 - BWG - incorrect explanation (body weight gain)

Re: “Body weight gain” was used to replace “BW gain” for the explanation. Please see line 204 in the revised manuscript.

6. Line 219 - it is not necessary to repeat formulas mentioned before in M&M

Re: The formulas was removed from the note in the revised manuscript.

7. How the terms of eggs quality evaluation were chosen?

Re: Egg quality including egg weight, yolk weight, yolk percentage, egg shape index, eggshell strength which are important for the indigenous breed were chosen. These traits are also the traits which may potentially affected by light environment. We add some words in line 167 in the revised manuscript.

8. What was the duration of experiment?

Re: The duration was the whole birds’ rearing period from day-old to 43 week. This information was added in line 112-113 in the revised manuscript.

Reviewer 5 Report

Edits are in the attachment.

Author Response

Response to Reviewer 5 Comments

1. Line 19, “fowls” should be “fowl”.

Re: Done as suggested.

2. Line 24, Rewrite the sentence “which permit the design 24 to output different and complex light spectrum as required” as “which permit the design of output difference and complex light spectrums as required”

Re: It was rewritten as follows “which permit the design of different and complex light spectrums output as required”. Please see line 25-26 in the revised manuscript.

3. Line 25, line 34, “spectrum” should be “spectrums”.

Re: Done as suggested.

4. Line 28, “difference” should be “differences”.

Re: Done as suggested.

5. Line 53-54, Rewrite the sentence “which are of luminous efficient and long lifespan” as “which are luminous efficient and have long lifespans”

Re: Done as suggested.

6. Line 55-56, “Extensive researches have been” should be “Extensive research has been”

Re: Done as suggested.

7. Line 337-338, There should be some response in the conclusions related to the CFL used in the study. “This study examined the effects of three designed LED lights with different defined spectral proportion on growth” should be “This study examined the effects of three designed LED lights with different defined spectral proportion and one CFL on growth”

Re: Done as suggested.